# Contemporary Developments and Emerging Trends in the Application of Spectroscopy Techniques: A Particular Reference to Coconut (*Cocos nucifera* L.)

**DOI:** 10.3390/molecules27103250

**Published:** 2022-05-19

**Authors:** Ravi Pandiselvam, Rathnakumar Kaavya, Sergio I. Martinez Monteagudo, V. Divya, Surangna Jain, Anandu Chandra Khanashyam, Anjineyulu Kothakota, V. Arun Prasath, S. V. Ramesh, N. U. Sruthi, Manoj Kumar, M. R. Manikantan, Chinnaraja Ashok Kumar, Amin Mousavi Khaneghah, Daniel Cozzolino

**Affiliations:** 1Physiology, Biochemistry and Post-Harvest Technology Division, ICAR-Central Plantation Crops Research Institute, Kasaragod 671124, Kerala, India; s.v.ramesh@gmail.com; 2Dairy and Food Science Department, South Dakota State University, Brookings, SD 57007, USA; martinez.monteagudo@gmail.com; 3Department of Family and Consumer Sciences, New Mexico State University, Las Cruces, NM 88003, USA; 4Chemical & Materials Engineering Department, New Mexico State University, Las Cruces, NM 88003, USA; 5School of BioSciences and Technology, VIT University, Vellore 632014, Tamil Nadu, India; divsarivan@gmail.com; 6Department of Biotechnology, Mahidol University, Bangkok 12120, Thailand; surangnajn8@gmail.com; 7Department of Food Science and Technology, Kasetsart University, Bangkok 10900, Thailand; khanashyamac@gmail.com; 8Agro-Processing & Technology Division, CSIR-National Institute for Interdisciplinary Science and Technology (NIIST), Trivandrum 695019, Kerala, India; kothakotaanjanikumar23@gmail.com; 9Department of Food Process Engineering, NIT, Rourkela 769008, Odisha, India; arun16foodengg@gmail.com; 10Agricultural and Food Engineering Department, Indian Institute of Technology Kharagpur, Kharagpur 721302, West Bengal, India; sruthisparks@gmail.com; 11Chemical and Biochemical Processing Division, ICAR-Central Institute for Research on Cotton Technology, Mumbai 400019, Maharashtra, India; manojkumarpuniya114@gmail.com; 12Department of Food Safety and Quality Assurance, College of Food and Dairy Technology, Chennai 600051, Tamil Nadu, India; ashokkumarchinnaraja@gmail.com; 13Department of Food Science and Nutrition, Faculty of Food Engineering, University of Campinas (UNICAMP), Campinas 13083-875, SP, Brazil; 14Department of Fruit and Vegetable Product Technology, Prof. Wacław Dąbrowski Institute of Agricultural and Food Biotechnology, 02-532 Warsaw, Poland; 15Centre for Nutrition and Food Sciences, Queensland Alliance for Agriculture and Food Innovation (QAAFI), The University of Queensland, Brisbane 4072, Australia

**Keywords:** coconut, FT-NIR-based technique, oxidation, peroxide value, virgin coconut oil, tender coconut water

## Abstract

The number of food frauds in coconut-based products is increasing due to higher consumer demands for these products. Rising health consciousness, public awareness and increased concerns about food safety and quality have made authorities and various other certifying agencies focus more on the authentication of coconut products. As the conventional techniques for determining the quality attributes of coconut are destructive and time-consuming, non-destructive testing methods which are accurate, rapid, and easy to perform with no detrimental sampling methods are currently gaining importance. Spectroscopic methods such as nuclear magnetic resonance (NMR), infrared (IR)spectroscopy, mid-infrared (MIR)spectroscopy, near-infrared (NIR) spectroscopy, ultraviolet-visible (UV-VIS) spectroscopy, fluorescence spectroscopy, Fourier-transform infrared spectroscopy (FTIR), and Raman spectroscopy (RS) are gaining in importance for determining the oxidative stability of coconut oil, the adulteration of oils, and the detection of harmful additives, pathogens, and toxins in coconut products and are also employed in deducing the interactions in food constituents, and microbial contaminations. The objective of this review is to provide a comprehensive analysis on the various spectroscopic techniques along with different chemometric approaches for the successful authentication and quality determination of coconut products. The manuscript was prepared by analyzing and compiling the articles that were collected from various databases such as PubMed, Google Scholar, Scopus and ScienceDirect. The spectroscopic techniques in combination with chemometrics were shown to be successful in the authentication of coconut products. RS and NMR spectroscopy techniques proved their utility and accuracy in assessing the changes in coconut oil’s chemical and viscosity profile. FTIR spectroscopy was successfully utilized to analyze the oxidation levels and determine the authenticity of coconut oils. An FT-NIR-based analysis of various coconut samples confirmed the acceptable levels of accuracy in prediction. These non-destructive methods of spectroscopy offer a broad spectrum of applications in food processing industries to detect adulterants. Moreover, the combined chemometrics and spectroscopy detection method is a versatile and accurate measurement for adulterant identification.

## 1. Introduction

Coconut (*Cocos nucifera* Linnaeus) belongs to Palmae and the subfamily *Cocoideae* and is a perennial palm prominently cultivated in coastal and island ecosystems [1]. The countries that top the list in the global production and consumption of coconut are Indonesia, Philippines, and India, followed by Sri Lanka, Brazil, Vietnam, Thailand, Papua New Guinea, Mexico, and the Pacific Islands [2]. About 50 percent of unprocessed, unfinished, and value-added coconut products are intruded globally, mainly from the Asian and Pacific countries [3]. Sri Lanka and Indonesia produce over 90% of the total share of coconut milk/cream. Simultaneously, over 50% of the market for desiccated coconut powder is dominated by Indonesia and the Philippines, according to the Asian and Pacific Coconut Community (APCC) in 2015. India is the leading producer of coir and coir products such as yarn, mats, and rugs, providing employment and empowerment for women [4]. With the rising market demands, the coconut processing sector is progressing, with more diversification, and enhanced production and productivity. Figure 1 represents various processing and the products obtained from the coconut.

Coconut is one of the richest natural plant sources of nutritional and medicinal components with broad applications in treating tumors, inflammation, and attacks of microbes and insects [5]. The endosperm forms the edible portion of the coconut fruit that includes coconut meat and coconut water [6]. The principal biochemical components of coconut water arecytokinins (e.g., kinetin and trans-zeatin), which exhibit actions against aging, carcinogens, and anti-thrombotic effects [7]. The byproducts of coconut meat, coconut milk, coconut cream, desiccated coconut powder, and coconut oil are extracted from coconut. Copra, the dried form of coconut meat, is heat-processed, and coconut oil is extracted using an expeller. Coconut oil contains medium-chain fatty acids (MCFAs), especially capric, caproic, lauric, and caprylic acids, readily absorbed without being stored as cholesterol. Coconut oil in the diet improves the high-density lipoprotein (HDL-c), which is good cholesterol that provides cardio-protectant effects [8]. Virgin coconut oil (VCO) is a variant of coconut oil extracted from coconut milk without direct heat through centrifugation and fermentation (Manikantan et al., 2018). Coconut milk, an oil–protein–water emulsion, is used in many cuisines and is a healthy alternative to milk for lactose intolerant vegans worldwide (Beegum et al., 2021). It also serves as a potential natural antioxidant that quenches the free radicals associated with deadly diseases such as cancer [9,10]. The concentrated form of coconut milk is called coconut cream and has similar applications as coconut milk. The meat is shredded, dried, and used as desiccated coconut or milled to desiccated coconut powder, and is applied in bakery and confectionery industries (Pandiselvam et al., 2022). The inedible portion of the fruit, such as the husk, is biodegradable and used in the manufacturing of activated carbon and charcoal in water purifiers and pharmaceutical industries [11]. The other value-added products from coconut include coconut honey, coconut sauce, nata de coco, coconut lemonade, coconut chips, coconut candy, coconut flour, coconut vinegar, coconut inflorescence sap, coconut sugar, coconut jaggery, and so on. Coconut and its derivatives are also used in nanoparticle synthesis for commercial applications (http://www.agritech.tnau.ac.in/access on 2 February 2022).

The quality evaluation of coconut and its products is of the utmost importance in nutrition and safety. The physical and chemical attributes include size, shape, density, moisture, fat, texture, hardness, color, and deformed surfaces that define the product’s quality [12]. The conventional testing methods involve sample preparation techniques that destroy their original nature and, more importantly, consume more time and human resources (Rifna et al., 2022; Pandiselvam et al., 2021; Kaavya et al., 2020). The titrimetric methods are sensitive; however, they require more time for sampling and analysis, are highly expensive and unreliable, need more resources, require trained personnel, have a high probability of error, and, more importantly, detrimental sampling cannot be used in line with processing. The significant advantages of non-destructive analysis are its reproducibility, reliability, ability to conduct any matrix analysis without injury to the samples, real-time processing, it is rapid and accurate, requires fewer human resources, has a better detection limit even at the micro-levels, and enables the analysis of diverse parameters [13,14].

The food products’ optical properties are considered while designing non-destructive testing equipment by utilizing the characteristic absorption spectra for the components of interest [15,16,17,18]. Advanced sensing methods such as optical, mechanical, spectroscopic, acoustic, E-nose and E-tongue methods are utilized for quality detection. The non-destructive spectroscopic techniques such as nuclear magnetic resonance and dielectric spectroscopy that employ radio frequency, UV–VIS spectroscopy, IR Spectroscopy, RS, FTIR spectroscopy, and spectroscopic imaging such as hyperspectral imaging (HSI), have been employed for various food products [19,20].

For many years, spectroscopic methods have been in vogue for the quality evaluation of foods due to their advantages over conventional methods. These techniques consider generating, quantifying, and interpreting spectra ascending from electromagnetic energy interactions with the sample [21]. Many analytical experiments are being conducted using different spectroscopic methods.

The methods’ specificities differ with the sample studied, the type of interaction to be monitored, and the zone of the electromagnetic spectrum employed. The IR, NIR spectroscopy, and NMR techniques are employed for analyzing the nutritional compositions of raw material and final products, whereas UV–VIS spectroscopy, fluorescence, MIR and RS are employed in the monitoring of food quality [22,23].

UV–VIS spectroscopy is based on Lambert–Beer’s law for estimating quality parameters such as anisidine value, peroxide value, and total oxidation value indicators of oxidation oils. The principle behind Lambert–Beer’s law relies on the ray of parallel monochromatic light, which illuminates the tested matrix [24]. After passing through the sample of a specified thickness, the light of a specific wavelength gets absorbed using the electron movement from the ground state to an excited state, reducing the amount of transmitted light. The processing of UV spectrum data consists of four steps: preprocessing spectrum data, extracting the characteristic wavelength, establishing model parameters, and evaluating model performance. Fluorescence spectroscopy is a susceptible and selective technique that is efficiently used to detect food-borne pathogens, fungal toxins, harmful additives, protein changes, and adulteration in oils [25]. Two basic types of spectra are generally assessed in conventional fluorescence spectroscopy. An emission spectrum is produced when a matter gets excited at a fixed wavelength, and the emission intensity is a function of emission wavelength. The excitation spectra can be obtained when the emission is set at a fixed wavelength. The emission spectrum is of importance in food analysis. A fluorescence excitation-emission matrix is obtained on a pair of spectra and is recorded at a different wavelength. A three-dimensional landscape can offer the intensity profile of a samples’ broad fluorescence over a wide range of excitation and emission wavelengths. Atomic absorption spectroscopy [26] is a widely used method for routine analytical control in different food products. The technique uses a sampling unit, a signal generation source, a spectrometer for sorting element-specific signals from the spectrum, and a detection system [27]. Recent developments in this field include classic flame systems, furnace systems, atomic fluorescence, and atomic emission spectrometry with plasma, as well as laser-plasma and glow discharges.

Vibrational spectroscopy is a promising spectroscopic technique that probes the intermolecular vibrations and rotations when the sample interacts with light. The primary optical techniques related to vibrational spectroscopy are Raman spectroscopy and infrared spectroscopy. The spontaneous inelastic scattering of light following the interaction with the sample is called Raman scattering. Elastic and inelastic scattering occurs on the photon and sample. The elastic scattering causes no changes to the energy levels of the molecules and therefore gives no molecular information. The elastic scattering is called Rayleigh scattering. In inelastic scattering, the photons lose energy to the molecule as vibrational energy known as Raman Stokes scattering. The molecular bonds present in the molecule provide different forms of vibration corresponding to the energy and thus provide the molecule’s fingerprint. The principle underlying IR spectroscopy involves the responses caused by the vibration of atoms at a particular spectrum divided into three bands, namely NIR, MIR, and far-infrared (FIR) with wavelengths of 0.7–5 μm, 5–40 μm, and 40–350 μm, respectively. Photons in the frequency of the IR region do not have enough energy to excite the electrons but cause vibrational excitations to the molecules depending on the atom and molecular groups present. NIR and MIR are used in the quantitative analysis of foods, of which the latter in combination with FTIR is employed to determine the structural-functional relationship of foods, the oxidative stability of various oils, and their adulteration [23,28,29,30,31]. Each of these can deliver well-defined conformational information for structural analysis. The NIR region has distinctive patterns that reveal the physicochemical compositions of the sample under study [32]. The responses and higher vibrations of NIR spectra deal with the molecular bonds, which cause a lesser absorption of NIR light in the organic materials, but the incidence of these forbidden transitions renders this region unique from other methods. Comparing NIR with IR spectroscopy, IR quantifies the specimen’s fundamental vibrations, which are very sensitive to the investigated compounds’ structure. There are two distinct regions under the MIR spectroscopy: the functional group and fingerprint regions. The former gives the most pertinent data to interpret the spectra, such as the X-H stretching, and C-H stretching in aldehydes, double-bonded functional groups, and triple bonds [33]. The latter is further complicated and overlapped but used to assess carbohydrates in various food products. The RS is used to quantify lipid components, detect adulteration, and monitor food interactions [34]. It works on the basic principle of scattering for photons and other molecules’ interactions. The Raman spectral information and chemometric methods help differentiate and classify the foods based on radiation absorption at different levels corresponding to specific wavelengths and other factors such as the process conditions and product composition [35]. Raman spectrum analysis usually consists of specific steps: sample preparation, the generation of the spectrograph, the preprocessing of spectral data, the extraction of the characteristic band, and establishing and applying classification or prediction models.

Chemometrics is a powerful tool that can be combined with spectroscopy to enhance and extract more valuable chemical data from the measurement data with the help of mathematical and statistical techniques. Calibration is one of the initial steps where chemometrics is applied in spectroscopy. Calibration is the process of correlating, modeling or relating the measured instrument responses (peaks, transmittance, absorbance, emission, retention time, peak area) with the concentration of the analyte. A multivariate calibration approach is usually applied in spectroscopy, as it considers the actual experimental data instead of individual signals [36]. Compared to using a single parameter, incorporating the complete spectral information results in superior prediction values. The spectroscopic study to determine the concentration of one or more of its constituents is a typical example of multivariate calibration in action. The Lambert–Beer law has been used in combination with K-matrix analysis, C-matrix analysis, principal component regression (PCR), partial least-square regression (PLS), and classical least squares (CLS) for the spectrophotometric response of concentration for a variety of applications. PLS is the most widely used calibration method in electroanalytical chemistry because of the quality of the calibration models created and the ease with which they may be implemented due to the availability of PLS software. In PLS, latent variables are created concurrently with the calibration model so that each latent variable is a linear combination of the original measurement variables rotated to provide the best possible correlation with the information supplied by the property variable. Generally, a preliminary exploratory analysis is performed to determine the reproducibility of the measurements. The most often utilized approaches for this purpose are PCA and clustering. CA is a dimension reduction approach that generates a small number of new variables known as principal components (PCs) from linear combinations of the original variables. Classification models are further used for discriminating the problem samples, where delimiters are defined between established classes so that the problem samples are permanently assigned to one of the established classes. Linear discriminant analysis (LDA), partial least squares discriminant analysis (PLS-DA), k-nearest neighbors (kNN), and support vector machines (SVM)are commonly used for this purpose.

Another critical application of chemometrics in spectroscopy is during sample selection. Any selection strategy seeking to eliminate redundancy in sample populations intended for calibration or validation is included in the broad definition of sample selection strategies for spectroscopic analysis. The best sample selection approaches would limit sample populations to the bare minimum of samples required to reflect all relevant spectral variance. As fewer samples are required for calibration, sample selection improves robustness. The selection of sample sets based on the statistical evaluation depends on the sample’s spectral similarity compared to a whole sample set or a subset of a sample set. To determine this similarity between different spectra, chemometric tools such as root mean square deviation (RMSD) or correlation (r) or the coefficient of determination (R^2^) are used.

In recent years, there has been increased interest in the potential application of spectroscopic methods in the food industry. Since consumers today are more aware of the health benefits of coconut and coconut-based products such as VCO and tender coconut water, the market demand for these products has recently been increasing. This has led to an increase in food frauds, and the surge in consumer demand for a consistent supply of safe, healthy, and traceable food has caused the processing industry to shift to faster, more reliable quality control and agricultural product assessment methods. Spectroscopic methods are of boundless advantage to modern-day scientific research with rapid and less time-consuming processing steps at a comparatively low cost. In this context, the purpose of this study is to present an overview of recent research on the use of non-destructive testing methods in coconut quality evaluation, such as NMR, IR spectroscopy, UV–VIS spectroscopy, fluorescence spectroscopy, FTIR, and Raman spectroscopic techniques in a comprehensive manner. The current review also aims to elucidate the applications of the spectroscopy technique with chemometric principles and how it distinctively evaluates various coconut products’ quality characteristics to reach their authenticated consumer market.

## 2. Methods

This study aims to provide a systemic review of spectroscopic techniques and their application in quality evaluation and adulteration detection in coconut-based products. For the development of the manuscript, published articles were collected from internet sources, mainly from databases such as PubMed, Google Scholar, Scopus and ScienceDirect with key words including spectroscopy, coconut adulteration, coconut water adulteration, VCO adulteration, NMR, FTIR, IR, UV, coconut quality, chemometrics, etc., and in total about 140 articles were collected and around 130 articles published after the year 2000 were short listed. Articles from before 2015 were selected due to their relevance to the selected topic and due to a lack of recent studies. The selected articles were thoroughly studied and critically analyzed for the preparation of the manuscript.

## 3. Results

### 3.1. Application of Spectroscopy Techniques for Quality Evaluation

Recent IR spectral analysis applications in the quality evaluation of different coconut products are highlighted in Table 1. The application of spectroscopic techniques for individual coconut products is presented in this section.

#### 3.1.1. Coconut Oil

Coconut oil has about 64% of the MCFAs that prevent various ailments and prevent the build-up of bad cholesterol, in contrast toother oils [55,56]. It is crucial to measure the physicochemical characteristics such as acid value, free fatty acid content, anisidine value, peroxide value, and the color of the oils to establish their quality [57]. Applications of non-invasive spectroscopic techniques are being explored for evaluating the quality of oils for rapid and online applications. The primary spectral measurement of coconut products is shown in Figure 2. Both RS and NMR spectroscopy techniques have proven their utility and accuracy in assessing changes in chemical and viscosity profiles [58]. FTIR spectroscopy coupled with IR was utilized to analyze the oxidation levels and determine the oils’ authenticity [58,59]. The authors Dayrit et al. [60] studied the lipid profile of coconut oil using NMR spectroscopy and established that the results of free fatty acid estimation were comparable to that of titrimetric methods. It was concluded that NMR was an effective method for analyzing fats and other constituents and differentiating the oils. The authors of paper [61] characterized the lipid profile of coconut oils by electrospray ionization mass spectrometry (ESI-MS) to distinguish the origin and authenticity of coconut and VCO from other edible oils.

#### 3.1.2. Coconut Milk

Coconut milk emits oil in water extracted from coconut meat by grinding it with or without water [62]. The primary composition of coconut milk is fat and water and other components such as carbohydrates, protein, and minerals. Coconut milk is used for making soups, curries, or packed as such and is made commercially available. The milk’s fat content is responsible for its mellow taste, and it not only accords the flavor but also determines the milk’s ability to maintain its quality. Hence, it is pertinent to verify the fat content to evaluate the quality of the product, an determine whether there is a need for the development of rapid methods which could be used in line with the processing conditions. Based on Zhu et al.’s [63] investigations, it was deduced that the reflection spectrum of coconut milk showed peaks at 1060nm and 1250nm while using a UV-VIS-NIR spectrophotometer. A Y-type optical fiber measurement system was used to measure fat, which can be employed rapidly for a real-time measurement with high accuracy and convenience. Cheevitsopon and Sirisomboon [44] also explored the feasibility of employing NIR spectroscopy to estimate coconut milk’s fat content in curry soups. A FT-NIR-based analysis of various samples confirmed the acceptable levels of accuracy in prediction. The long-chain fatty acid moieties influence the measurements, and further validation is required for application in a wide range of measurements during online processing.

Further studies on the fat and moisture content measurement with NIR in coconut milk by Wattanapahu et al. [64] proved that NIR is more convenient, requiring little sample preparation time and an accurate analytical tool. The samples measured at 400–2500 nm and 0.5 nm intervals yielded accurate measurements of fat and moisture contents correlated with coconut milk quality. Another important indication of coconut milk’s quality is its pH, as it determines the consumer acceptability and the quantum of food spoilage. Thitibunjan and Sirisomboon [45] used rapid, convenient, and accurate NIR spectroscopy for the estimation of the pH of curry soup containing coconut milk in diffuse reflection mode at a wave number of 12,500–3600 cm^−1^. The samples were analyzed with an FT-NIR spectrometer at specific conditions, and the results were validated with the actual pH values. Table 2 gives a brief synopsis of different studies that attempted to analyze coconut milk constituents using spectroscopy. These studies have suggested that NIR could be a proper alternative method to measure instant soups’ pH and total solids with coconut milk as the primary ingredient in the production lines for process control and quality assurance [65]. Thus, spectroscopic methods have proved to be more rapid and accurate. However, further studies are warranted to validate the correlation of physicochemical properties with the products’ deterioration and the predicted results.

#### 3.1.3. Tender Coconut Water

Coconut water is considered a refreshing drink with a sweet taste and therapeutic benefits. However, it is prone to rapid deterioration due to different microbial and physiochemical processes [38] that severely limit its marketability. Hence, it is necessary to assess its quality during its processing and storage. Furthermore, coconut water’s biochemical features vary with the different genotypes of coconut and with different stages of maturity. Hence, it is mandatory to evaluate and monitor the quality of coconut water. Previously, classical analytical techniques were used for quality evaluation, including monitoring the physicochemical parameters of coconut water such as turbidity, soluble solids, titratable acidity, polyphenol activity and oxidase enzymes [38,68].

Nevertheless, currently, many advanced and instrumental methods are being used for analyzing and evaluating the quality and authenticity of coconut water [39,69,70]. Coconut water also contains many relatively poorly investigated and unknown solutes with special biological effects. The development of more advanced detection and spectroscopy methods can also detect these novel compounds in this context.

Fourier transform ion cyclotron resonance mass spectrometry (FT-ICR MS) is a method with high accuracy and resolution that allows for the easy characterization of complex organic mixtures. All this can be performed without prior extraction and isolation techniques. The FT-ICR MS coupled with an electrospray ionization (ESI) source helps detect and identify chemical compounds synthesized when coconut water undergoes natural aging [37]. Three days of aging resulted in an evident change in the chemical profile with the production of different compounds such as citric acid, galacturonic acid, gluconic acid, and saccharic acid, which was attributed to the hydrolysis reactions that convert polysaccharides into oligosaccharides and monosaccharides, and disaccharides into monosaccharides, followed by oxidative degradation. In fresh coconut water, it was deduced that FT-ICR MS helped characterize different molecules such as sucrose, glucose, fructose, and gluconic acid that are considered natural identification markers for predicting the quality of coconut water.

Proton nuclear magnetic resonance (1H NMR) is an analytical tool to analyze complex mixtures and determine different fruit juices and beverages [71,72].Proton nuclear magnetic resonance (1H NMR), along with FTIR and GC-MS, has been investigated for evaluating the quality of coconut water following the addition of L-ascorbic acid to reduce free fatty acid synthesis [38].NMR spectroscopy with chemometrics was also employed to assess variations in the chemical compositions and primary metabolites of coconut water when subjected to different processing treatments [39]. It was observed that NMR and chemometrics provided quick and non-destructive quantitative information about primary metabolites found in coconut water (processed and unprocessed). Additionally, these methodologies require significantly less quantum of samples for initial analysis.

NIR spectroscopy (NIR) efficiency was investigated to evaluate the post-harvest quality of coconut water [40]. It was a comprehensible tool to monitor coconut water deterioration during the different stages of maturity. Another robust analysis and authentication method employed is stable isotope ratio mass spectrometry (SIRMS). SIRMS was used to evaluate and detect undesired additives such as C-4 plant sugars of cane sugar and maize syrup origin in coconut water [41]. RS with chemometrics has also been employed to assess sugar additives in fresh coconut water, which affects their overall quality [42].

These different studies have demonstrated the potential of advanced spectroscopy methods in the quality evaluation of coconut water. However, their utility requires further exploration to analyze the different biochemical changes in the coconut water and modulations in flavor components throughout storage. Their application for the quality evaluation of coconut water as a function of the coconut genotype, soil component variation and industrial conservation method also requires investigation.

#### 3.1.4. Virgin Coconut Oil

VCO is an emerging functional food owing to its numerous health benefits, such as reducing cholesterol and triglyceride levels, inhibiting the oxidation of low-density lipoproteins, and increasing the number of antioxidant components in our body [73]. They are also known to comprise a large quantum of MCFAs *viz*., capric, caproic, and caprylic acids, and are known to possess antimicrobial and antiviral attributes [46,74]. However, fats and oils are prone to deterioration during storage due to lipid oxidation. Due to lipid oxidation, hydroperoxides are formed, causing shorter breakdown molecules, including ketones, alcohols, carboxylic acids, and aldehydes [75]. These volatile products and toxic components cause the rancid flavor in oil, make the foods prepared using this rancid oil unappetizing and severely reduce their nutritional value; the final food products that remain are largely unacceptable to consumers [76,77]. Therefore, evaluating the oxidative stability during VCO storage and heating is essential for various food applications.

Various methods are being employed to measure the degree of the oxidative deterioration of edible oils, which focuses on quantifying the concentration of oxidation products (primary and secondary) and low molecular weight breakdown products of unsaturated fatty acids [78,79]. In this backdrop, FTIR spectroscopy is an ideal and accurate analytical technique that helps to determine the oxidative deterioration of VCO.

The biochemical changes in VCO during thermal oxidation were studied by FTIR [78]. A prominent FTIR peak at 1739 cm^−1^ indicated carboxylic compounds in the oil during its oxidation. Additionally, the absence of peaks at 3300 cm^−1^ suggested that hydroperoxides or free fatty acids were not synthesized during the process of thermal oxidation. FTIR spectroscopy was also studied along with attenuated total reflectance (ATR) for evaluating the peroxide value of VCO [47]. The actual peroxide value was quantified following the standard AOCS method and a predicted FTIR analysis value. This was followed by the calibration and validation of the models, wherein the R^2^ value of 0.983 was obtained. The stability analysis of VCO during its prolonged deep frying of 8 h using FTIR spectroscopy [48] revealed that VCO was stable since the peak of 1739 cm^−1^ corresponding to carboxylic acids was conspicuously missing.

FTIR spectroscopy can monitor the different biochemical parameters of edible oils, including VCO. It is considered a green technology and a fingerprint method since a relatively minor quantity of chemicals and reagents are required for the analysis [79,80]. Zicker [49] used FTIR to evaluate the quality of VCOs. Generally, the acid–base titration method is used as a reference method in oils to determine the presence of free fatty acids, but it involves utilizing toxic and flammable solvents, consumes a lot of time, and is prone to error [81]. Zicker et al. [49] reported FTIR spectroscopy to be beneficial over the acid–base titration method due to its non-destructible nature, shorter time, and simplified analysis technique.

Furthermore, a statistical technique, chemometrics, could be well combined with FTIR for the quantitative analysis of the free fatty acids of VCO and in monitoring their quality parameters [49,79]. FTIR spectra’s complexity is interpreted using chemometrics’ application involving different methods (mathematical and statistical) to extract valuable knowledge from the spectral data. RS, along with chemometrics, was analyzed to evaluate the thermal stability of VCO along with other edible oils [50]. These methods were found to be entirely accurate and fast, suggesting these methods could be successfully used in commercial establishments such as restaurants for evaluating the standards of different edible oils.

Scientific literature recommends the application of FTIR spectroscopy for evaluating the quality of edible oils. However, studies on the quality evaluation of VCO for commercial applications are scarce, whereas the available studies mainly focus on detecting adulteration in VCO or evaluating their oxidative stability.

#### 3.1.5. Edible Coconut Products

Under field conditions, the coconut crop is prone to absorbing and accumulating heavy metals, like many other crops. The presence of heavy metals such as lead, cadmium, and mercury in the products derived from coconut are reported to be toxic when present in excess amounts. They can then contribute to several health disorders in human beings [43]. Therefore, heavy metal absorption affects the quality of products derived from coconuts, such as fresh coconuts, coconut cream, coconut milk powder, and coconut-based curries. However, applications of spectroscopy-based analytical methods have made it possible to detect heavy metals even if they are present in a minuscule proportion.

A flame atomic absorption spectroscopy (FAAS)-based analysis assessed the quantum of heavy metals in different coconut products [43]. However, this study could not detect lead, arsenic, cadmium, aluminum, and chromium due to the lesser detection limit of FAAS. The minimum detection level of the atomic absorption spectrophotometer for lead, arsenic, cadmium, and aluminum has been reported to be 0.5 mg/kg, and for chromium it is 1.5 mg/kg [43], and below this level, these heavy metals cannot be detected by FAAS. Hence, an analytical method with more sensitivity is required for the accurate quality evaluation of the coconut products.

Coconut curry soup is another coconut-based product that is popular worldwide. It is made from coconut milk and contains many saturated fats [82]. Therefore, curry soups’ fat content must be evaluated to match the final product’s standard. The NIRmethod was studied to analyze red, green, massaman, and Panang curry coconut soups [44]. It was concluded that NIR could be used in factories producing coconut curry soups with complete accuracy. The NIR method was also evaluated to study different coconut curry soups’ pH to analyze their deterioration [45]. This method was found to have acceptable prediction accuracy. The NIR method has also been utilized to evaluate the total solids in coconut curry soups [65]. Total solids analysis is an important quality parameter as it determines the preservation and stability of the product.

Interestingly NIR-based analysis was found to have good accuracy for total solids measurement. To date, NIR has been the mainstay for analyzing coconut products [83]. However, more studies are required for the quality evaluation of coconut-based products utilizing various suitable spectroscopy methods. Furthermore, more accuracy is imperative for detecting the low concentrations of heavy metals in coconut-based products for better-quality evaluation.

#### 3.1.6. Non-Edible Coconut Products

The quality evaluation of non-edible coconut products such as coconut shells is equally important as they are prone to cracking due to climate variations [51]. Identifying cracks in shells is difficult as they are not observed externally. Traditionally, farmers have adopted the buoyancy method to evaluate cracked shells [51]. However, this method is tedious, as evaluating every coconut fruit in a bunch is a time-consuming process. Furthermore, if not performed correctly and cracked shells surface in the trimmed coconuts in the factories, those coconuts are rejected, affecting the growers financially.

NIR has been extensively applied to detect cracked shells in a coconut. This method has been used previously to detect internal defects in citrus fruits, moody cores in apples, and internal bruising in blueberries [84,85,86]. Noypitak, Imsabai, Noknoi, Karoojee, Terdwongworakul and Kobori [51] studied and evaluated NIR application and acoustic response for detecting cracked shells in Thai young coconuts. The NIR method demonstrated outstanding potential for identifying cracked shells in coconuts in bunches.

Coconut husks are by-products of coconut industries that are generally abandoned but are appealing due to their rich nutrients and polymeric structures of cellulose, hemicellulose, and lignin. The coconut husks can be used as a renewable resource in biorefineries to produce biofuels [87]. Biofuel yield from coconut husks depends on their quality and chemical composition, which were previously analyzed using expensive and time-consuming traditional methods. However, accurate NIR methods have been developed [52]. NIR and chemometrics were applied to evaluate the cellulosic lingo components of coconut husks based on their spectral data for biofuel production and were found to be robust [52]. FTIR spectroscopy was applied for coconut husks’ chemical characterization and their adhesive capacity and application in tannin extraction [53]. The tannin-specific characteristic peaks are observed in FTIR spectral data demonstrating the husks’ utility in the production of adhesives.

Coconut fibers are derivatives of coconut husks with broad applications in different agricultural, biotechnological, and industrial processes. However, it is crucial to evaluate their quality characteristics to develop high-quality final products. Brígida et al. [54] reported they chemically treated the coconut fibers under different methods to improve their physicochemical properties for possible novel applications. FTIR spectroscopy analysis was employed to evaluate modifications in their chemical composition, which could not be analyzed using the scanning electron microscope (SEM) technique.

Hence, it is evident that very few prominent investigations have been performed using spectroscopy techniques for the quality evaluation of non-edible coconut products. Hence, more studies are warranted to investigate the accuracy and effectiveness of other spectroscopy techniques for quality evaluation.

### 3.2. Application of Spectroscopy Techniques for Adulteration Detection/Authentication

The detection of adulterants in various coconut products using spectroscopy is depicted in Table 3. The salient findings of spectroscopy techniques for the detection of adulteration are presented in this section.

#### 3.2.1. Coconut Oil

Compared to other saturated fats such as animal fats, coconut oil is rich in small MCFAs and has nutraceutical and health benefits [8]. Due to its relatively high market price, adulterating coconut oil with other cheap lower-grade oils is common [22]. The addition of sesame, canola, sunflower, soybean, corn, peanut, castor bean, babassu, palm kernel, mineral, and Vaseline oils as an adulterant on coconut oil was analyzed using RS [89]. RS in combination with a chemometric tool, multivariate curve resolution–alternating least squares (MCR-ALS) analysis, was used in the quantitative and qualitative detection of adulterants in coconut oil. The MCR-ALS method analyzes the experimental data through a bilinear decomposition model to identify the mixture’s pure component. The differences in spectral intensities between coconut oil and adulterants were observed at 1264 and 1658 cm^−1^. The model successfully identified and quantified the adulterants in coconut oil at a concentration above 2%.

Munir, Musharraf, Sherazi, Malik, and Bhanger [88] developed a rapid detection method for detecting lard adulteration in coconut oil using FTIR. The FTIR data from the frequency region 1246.75–1078.01 cm^−1^ were used for analyses. Due to similar functional groups, coconut oil and lard spectra were almost similar. However, coconut oil does not reach a peak at 3010 cm^−1^, which is specific to the trans=CH_2_ stretching vibration, whereas a peak at 1100 cm^−1^ is visible for CO, differentiating the oil from the lard. Additionally, a peak at 750 cm^−1^ was smaller for coconut oil than the lard. The partial least square (PLS) algorithm developed had a correlation coefficient (R^2^) of 0.9577 and an RMSEC of 0.049.

#### 3.2.2. Virgin Coconut Oil

VCO is a functional oil derived from coconut milk either by physical or fermentation processes [97]. Due to its potential health benefits, VCO is gaining a lot of consumer acceptance [98,99]. VCO can be adulterated with low commercial value oils due to economic reasons. Man [91] has used FTIR spectroscopy to analyze the presence of canola oil in VCO. The fingerprint peaks of VCO were obtained at a frequency region of 1120–1655 cm^−1^. Compared to canola oil, VCO is rich in saturated fatty acids, and hence no peaks that correspond to Cis=C-H and Cis –C=C–stretching (at 3007 cm^−1^ and 1655 cm^−1^) were visible, which otherwise is specific for canola oil. The chemometric techniques PLS and discriminant analysis (DA) are applied to quantitatively determine the canola oil’s adulteration in VCO. For the quantification of canola oil on VCO, multivariate calibration models, PLS and principal component regression (PCR), were applied in the spectral region of 1200–900 and 3027–2985 cm^−1^, whereas for the discrimination of pure oil from the adulterated sample, the DA was used.

Similarly, FT-MIR spectroscopy coupled with chemometric techniques were used to identify corn (CO) and sunflower oil (SFO) adulterants in VCO [75]. The MIR spectra were recorded using a horizontal attenuated total reflectance attachment (HATR). Due to saturated bonds, VCO has not shown any peaks at 3008 cm^−1^. At the same time, in the fingerprint regions of 1120–650 cm^−1^, CO and SFO exhibited two peaks, and VCO showed a single peak. The VCO adulterated with CO was classified using the spectral frequencies in the range of 3028–2983, 2947–1887, and 1685–868 cm ^−1^. Frequencies between 3030–2980 cm^−1^ and 1300–1000 cm^−1^ were used to classify the VCO adulterated with SFO. The DA analysis was performed with the Coomans plot, which correctly detected the adulterant in VCO with ten principal components. The quantification was carried out with the help of the PLS algorithm. The FTIR values’ model correlation with the estimated values for both the adulterants (R^2^) was 0.999.

Another common adulterant in VCO is palm kern olein. An FTIR equipped with an ATR attachment analyzed the VCO and adulterated samples. As both PKO and VCO are lauric oils, with a lauric acid content of 47–48%, the oils showed almost super-imposable spectra for the naked eye. The prominent peaks were observed at 3100–280 cm^−1^, 1700–1800 cm^−1^ and 1400–900 cm^−1^. This frequency corresponds to C–H stretching, C=O stretching, and C–O–C stretching, respectively. However, the peak at 3006 cm^−1^ corresponding to cis C=CH stretching was absent in the spectrum of VCO, whereas the palm kernel oil presented a clear peak at 3006 cm^−1^ due to the presence of oleic acid. The PLS algorithm was calibrated with a good linear regression, with an R^2^ of 0.9875. The DS analysis was later applied in the wavelength region of 3500–700 cm^−1^ and could detect the palm kernel olein adulteration up to 1% in VCO [92].

FTIR spectroscopy was used to determine the level of adulteration in binary and ternary oil mixtures [67,100]. The binary mixture of VCO with olive oil (OO) and palm oil (PO) was analyzed with FTIR. The VCO and OO spectra showed differences at the frequencies 1111 cm^−1^ and 962 cm^−1^, which correspond to C-O stretching and CH=CH bending. The chemometrics analysis was performed using PLS and PCR techniques combined with a spectral range of 1120–1105 cm^−1^ and 965–960 cm^−1^. An R^2^ value of 0.999 was obtained for multivariate calibration models with PLS and PCR [67].

Similarly, for the VCO and PO mixtures, frequency ranges of 1120–1105 cm^−1^ and 965–960 cm^−1^ were used to calibrate PLS and PCR models, and R^2^ values of 0.999 and 0.974 were correspondingly obtained for each of the models, respectively.

In the case of ternary oil mixtures, VCO mixed with palm and olive oil was analyzed. A comparison of IR spectra of PO, OO, and VCO showed no bands in the frequencies of 3005 cm^−1^ and 1654 cm^−1^ for VCO, which were otherwise specific for the adulterants PO and OO. The frequencies at 3005 cm^−1^ and 1654 cm^−1^ correspond to cis =C–H and cis –C=C– vibrations, respectively, and indicate the presence of unsaturated fatty acids. The PLS and PCR calibration models were determined in the frequency region with the highest determination coefficient (R^2^). The study also found PLS to be a better prediction model than PCR for VCO ternary mixtures. The frequency range of 1200–1000 cm^−1^ for VCO ternary mixtures gave an R^2^ value of 0.999 for PLS [100].

A similar exercise was performed to determine the lard (LD) adulteration of VCO using FTIR. The major difference in VCO’s IR spectra was adulterated with lard, and pure VCO was the absence of peaks at 3006 cm^−1^ and 1098 cm^−1^ for the latter. The further quantification of lard in VCO was performed using PLS. The R^2^ between the predicted FTIR values with actual FTIR values was 0.99 at a frequency range of 3020–3000 cm^−1^ and 1120–1000 cm^−1^. Coomans’ plot was used for DA analysis, and it classified the lard based on six principal components. Lard was detected in VCO at a concentration of as low as 1% [93]. FTIR was also used to identify grape seed oil and soybean oil in a binary mixture with VCO [101]. The grape seed oil was detected in VCO by applying PLS calibration in the absorbance at 1200–900 cm^−1^ and 3027–2985 cm^−1^. To detect the presence of SO in VCO, PLS calibration was carried out using the combined wavelengths of 3025–2995 cm^−1^ and 1200–1000 cm^−1^.

FTIR-ATR was employed to detect the presence of paraffin oil (PFO) as an adulterant in VCO [90]. The spectral peaks in the regions of 3000–2800 cm^−1^ and 1800–700 cm^−1^ were considered the major informative spectral regions. As PO is entirely composed of n-alkanes (C_16_–C_20_), the two bands at 3000–2850 cm^−1^ and 1470–1450 cm^−1^ were due to –C–H stretching and –C–H bending. The PCA analysis was carried out in the spectral regions of 3000 to 2800 cm^−1^ and 1800 to 700 cm^−1^ to determine the resemblance and disparity between VCO and its PO adulterant. Linear discriminant analysis (LDA) was further used for the DA of the adulterant, and it detected up to 1% *v*/*v* of PO in VCO. The quantitative differentiation of PO in VCO was performed by applying statistical models such as PCR and PLS to provide a suitable calibration model based on the spectral information from the region of 1800–700 cm^−1^ characterized with a high R^2^ value of 0.99, the lowest RMSEP value.

#### 3.2.3. Tender Coconut Water

Due to the appreciable mineral content and health benefits, tender coconut water consumption is rapidly increasing every year [102,103]. Considering its high market demand and popularity, coconut water is a prime target for many malpractices such as adulteration. These adulterations mainly consist of three categories: stretching or bulking, where inexpensive adulterant is added to increase the volume or weight of the sample; masking, where the inferior quality product is masked by adding a foreign ingredient; and mislabeling, where an inferior quality product is labeled as a high-grade product. Spectroscopy techniques can be used to identify these adulterants in a non-invasive, fast, and efficient manner. One of the common adulterants in tender coconut water is milk, which is added to mask the enzymatic discoloration. The presence of bovine milk in fresh, tender coconut water was successfully detected using FTIR and RS [94]. As RS measures the inelastic scattering, the spectral measurements were affected by fat globules in the milk, which severely mask the signals at higher concentrations and remain a significant challenge. Compared to RS, FTIR has been proven to be a more powerful tool for detecting bovine milk adulteration in coconut water. The fingerprint peaks for bovine milk were observed at 2924 cm^−1^ and 2845 cm^−1^, representing the C-H vibrations of triglycerides, and the C=O ester bonds were observed at 1747 cm^−1^.

Additionally, bovine milk’s protein content gave peaks at 1655 cm^−1^ and 154 cm^−1^, corresponding to amide-I and amide-II bonds, respectively. Another major adulterant in coconut water is sugar, which increases the total soluble solid content. A recent study in Brazil used isotope ratio mass spectrometry (IRMS) to show that 65% of industrialized coconut water is adulterated with excess sugar [96]. Other productive spectroscopy techniques used to identify adulteration in coconut water are the combustion module–cavity ring-down spectroscopy (CM-CRDS) method and the low-cost alternative IRMS method, which are used on carbon isotopic (δ^13^) composition.

The CM-CRDS method distinguished up to ~5% sugar adulteration in coconut water (Picarro, 2017). A recent study has also demonstrated RS’s potential application in detecting simple sugars in coconut water. RS yielded discriminant peaks for fructose, glucose, and sucrose at 627 cm^−1^, 1123 cm^−1^, and 835 cm^−1^, respectively. RS detected the individual adulterant level even below 3% [42]. One of the most significant challenges in RS was its trouble in detecting the mixed sugar solutions. It was overcome by using a relatively sensitive proton NMR spectroscopy. Combining NMR spectroscopy with principal component analysis (PCA) and PLS regression developed a quantitative model with a sensitivity of as low as 1.3% adulteration [95].

### 3.3. Microbial Contaminants and Toxic Component (Heavy Metals) Detection in Coconut Products

Heavy metals have a density higher than 5 g·cm^−3^ and are also called trace elements because of their presence in very minute quantities in the environment [104]. Some heavy metals such as lead (Pb) and cadmium [105] are toxic to humans above a certain level, whereas some others such as copper (Cu), iron (Fe), and molybdenum (Mo) are called micronutrients which are toxic only if taken in high concentrations. A summary of different studies on quantifying heavy metals in edible coconut products using spectroscopic techniques is given in Table 4. AAS was used to study the amount of heavy metal contamination in coconut water in different studies (Xin, 2009). Eight heavy metals (Fe, Ni, Cu, Cd, Cr, Zn, Pb, and Se) were analyzed in fifteen coconut water samples collected from Dhaka, Bangladesh. The study showed many toxic heavy metals such as Ni, Cd, Cr, and Pb in coconut water. The average concentration of Pb in coconut water was 0.35 mg/L against the maximum permissible limit (MPL) of 0.1 mg/L. The content of Cr (10.98 mg/L) reported was 100 times higher than the MPL (0.1 mg/L). These highly toxic heavy metal contamination concentrations could be traced to the untreated industrial effluents contaminating the local environments. The study also reported a meager amount of trace nutrients such as Mg, Cu, and Zn in all tested samples.

The amount of heavy metals such as chromium (Cr), lead (Pb), iron (Fe), aluminum (Al), manganese (Mn), copper (Cu), cadmium [105], arsenic (As), and zinc (Zn) in fresh coconut kernel, coconut milk, coconut milk powder, and coconut cream from Sri Lanka was analyzed using AAS.67. The study showed varying Fe, Zn, Cu, and Mn in fresh coconut kernel, coconut milk, coconut cream, and coconut milk powder. The lowest quantities of Zn, Cu, and Mn were detected in the processed coconut products of coconut milk powder and coconut cream, which was likely due to the losses of these water-soluble minerals during processing. On the other hand, heavy metals such as Pb, Cd, As, Cr, and Al were present in concentrations lower than the minimum detection level of AAS. A study on Indian coconut water identified strontium (Sr), Mn, Cr, and Cd by atomic absorption spectrophotometry [106].

Another fast and accurate technique to analyze the heavy metal content in coconut water is the inductively coupled plasma (ICP) optical emission spectrometry (OES) method [107]. With an increase in plasma power, an increase in the sensitivity of the element analyzed was observed. However, ICP-OES with an optimized plasma power of 1.4 KW and nebulization flow rate of 0.5 L min^−1^ was used to determine the heavy metals in natural coconut water due to the increased noise. Ca, Mg, Mn, Fe, Zn, and Cu were determined at wavelengths of 317.9, 280.2, 257.6, 238.2, 213.8, and 324.7 nm, respectively. The limits of detection [107] (mg L^−1^) for various metals were finalized based on the radial configuration for Ca (0.06), Mg (0.004), Mn (0.02), and Fe (0.16) and axial configuration for Zn (0.008) and Cu (0.006). Staphylococcal food poisoning (SFP) is a common food-borne disease caused by staphylococcal enterotoxins (SE). Their detection in the food matrix remains a risk due to the lack of comprehensive immunological tools. A protein standard absolute quantification (PSAQ) technique for quantitative analysis of SE in coconut pearls was developed by Dupuis et al. [66] with a quadruple time-of-flight mass spectrometer (QTOF-MS).

## 4. Conclusions

Coconut products have gained more popularity due to their varied health benefits. The functional phytochemicals in the VCO have also led to various fraudulent activities using adulteration to reduce the cost. Although the biochemical characterization of the coconut products’ components is carried out using high-performance liquid chromatography (HPLC) and liquid chromatography–mass spectroscopy (LC-MS), the increasing popularity of the products and the practice of adulteration warrant the use of more robust techniques. The current advancements have offered many novel tools and equipment that can analyze the quality of various food products, including coconut products. The current review focuses on applying spectroscopic techniques to the quality evaluation of coconut products. Spectroscopy has been demonstrated as a rapid, nondestructive, and affordable detection technique that can replace older technologies, and thereby save on time, money, and experienced personnel. The combination of spectroscopy and chemometrics has also yielded compelling models that work even without rigorous sample preparation processes. The calibration and development of calibration models are the crucial and time-consuming steps that limit the application of spectroscopy for adulteration detection in industries, as it requires trained human resources expertise and arduous interpretation skills. However, once the calibration stage is accomplished, authentication can be carried out rapidly and cost-effectively with a single analysis. Moreover, there is an ever-increasing need for rapid, nondestructive technology for food authentication in the industry. However, the intrinsically complex properties of food cannot be predicted by using a single technique, and there is a need for developing various food profiling and fingerprinting techniques where the data from different technologies could be combined into a single matrix to come up with one effective model that provides the final prediction. The current review gives a comprehensive analysis on the various spectroscopic techniques that can be used for the authentication and analysis of various coconut-based products. The information provided in this review demonstrates the effectiveness of spectroscopic methods in the evaluation of coconut-based products. The review also explores various chemometric techniques, including calibration models and multivariate analysis methods that are used in various spectroscopic techniques. Most of these studies are purely academic. However, this review can be used as a reference for both industries and academicians who wish to develop prediction models for authenticating coconut-based products.

## Figures and Tables

**Figure 1 molecules-27-03250-f001:**
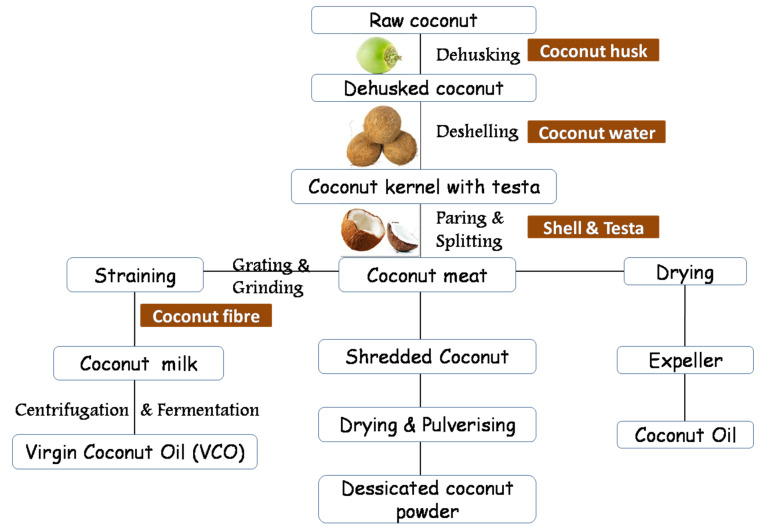
Steps involved in the processing of coconut.

**Figure 2 molecules-27-03250-f002:**
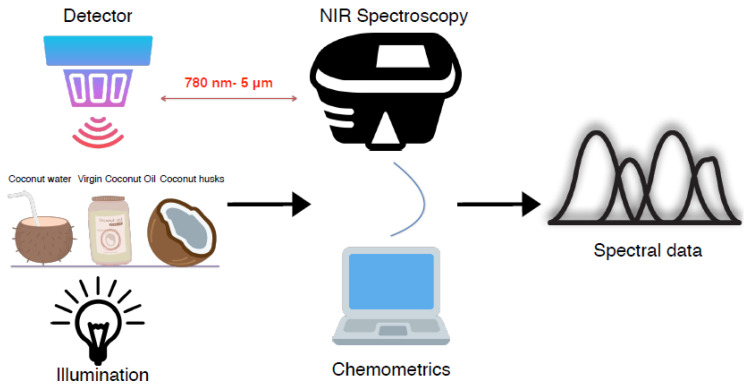
Schematic view of the spectral measurement of coconut products.

**Table 1 molecules-27-03250-t001:** Recent applications of infrared spectral analysis in the quality evaluation of different coconut products.

Coconut Product	No. of Samples	Spectral Analysis	Major Findings	Reference
Water from fresh and aged coconuts	7	FT-ICR mass spectrometry	Detection and identification of chemical compounds that are synthesized when coconut water undergoes natural aging.	[37]
Water from green coconuts with amaturation age of 5–7 months	8	1H NMR, FTIR (4000 to 400 cm^−1^) and GC-MS along with PCA statistical analysis	All these techniques were identified as rapid screening methods for the quantitative detection of micro filtered coconut water during storage.	[38]
Coconut water	12	NMR spectroscopy with chemometrics	Provide quick and non-destructive quantitative information about primary metabolites present in both processed and unprocessed coconut water.	[39]
Coconut water	192	NIR	Found to be a highly sensitive tool that helped to monitor coconut water deterioration during different stages of maturity.	[40]
Coconut water	54	Stable isotope ratio mass spectrometry	Detected the presence of added C-4 plant sugars such as cane sugars and maize syrups.	[41]
Fresh coconut water	155	RS with chemometrics	Accurate analytical method for the detection of added sugars in coconut water.	[42]
Coconut kernel, milk, milk powder and cream	44	FAAS	Detected the presence of heavy metals in the different coconut products.	[43]
Coconut curry soup	12	NIR (3600–12,500 cm^−1)^	NIR spectroscopy can be considered for use in factories producing coconut curry soups.	[44]
Coconut curry soup	73	NIR (3600–12,500 cm^−1)^	NIR spectroscopy can be used as an alternative method to evaluate the pH of curry soups.	[45]
Coconut curry soup	12	NIR (3600–12,500 cm^−1)^	NIR spectroscopy could be applied for the quality assurance of instant curry soups.	[44]
Virgin coconut oil	36	FTIR (4000–400 cm^−1^)	FTIR spectroscopy was able to detect carbonylic compounds from hydroperoxide decompositions.	[46]
Virgin coconut oil	30	ATR-FTIR (4000–650 cm^−1^)	Detection of peroxide values in virgin coconut oil.	[47]
Virgin coconut oil	8	FTIR (4000–500 cm^−1^)	FTIR spectra found the thermo-stability of virgin coconut oil samples even after 8h of frying.	[48]
Virgin coconut oil	72	FTIR (3100–680 cm^−1^) with PLSR	FTIR was found to be superior to the acid–base titration method for determining free fatty acids.	[49]
Virgin coconut oil	8	RS with chemometrics	RS could be used in restaurants for monitoring the quality of different oils.	[50]
Young coconut fruits	202	NIR (11,100–3996 cm^−1^) with acoustic response	Identified cracked shells in young coconuts remain in bunches.	[51]
Coconut husks	54	NIR with chemometrics	Evaluated the lignocellulosic components of coconut husks.	[52]
Coconut husks	4	FTIR spectroscopy (4000–400 cm^−1^)	Determined the adhesive capacity of coconut husks and their application in tannin extraction.	[53]
Coconut fibers	4	FTIR spectroscopy (4000–500 cm^−1^)	Evaluated modifications in the chemical composition of coconut fibers.	[54]

**Table 2 molecules-27-03250-t002:** Quantitative analysis of coconut milk constituents using spectroscopic techniques.

Spectral Analysis	Statistical Tool	Salient Findings	Reference
ESI-MS	PCA	Skim milk and milk sediment showed the highest match score for 7Sglobulin Glutelin OS with a very low score was identified for alkaline protein extract	[37]
ICP-OES	Factorial and Doehlert design	Direct determination of micronutrient minerals Predominant element was K, followed by Na, P, Mg and Ca	[66]
FT-IR	PLS	Presence of the carbohydrate source and water were detected Absence of protein and fat peaks	[67]
FT-NIR	PLS	Presence of vibration bands of CH_2_, indicating long-chain fatty acid moiety Fat content of milk highly affected the prediction of fat content in curry soup	[44]
FT-NIR	PLS	Presence of vibration bands of amides, hydrocarbons, and aliphatics NIR is a better alternative for evaluating total solids	[44]

**Table 3 molecules-27-03250-t003:** Detection of adulterants in various coconut products using spectroscopy.

Product	Adulterant	Spectral Analysis	Statistical Tool and Accuracy	Reference
Coconut oil	Lard	FT-IR (4000–400 cm^−1^)	PLS R^2^: 0.9577, RMSEC: 0.0488	[88]
Coconut oil	Sunflower, soybean, canola, sesame, corn, castor bean, peanut, palm kernel, babassu, mineral, and Vaseline oils	RS (3200–200 cm^−1^)	Multivariate curve resolution–alternating least squares (MCR-ALS) R^2^: 0.962–0.992, RMSEC: 916–2.944	[89]
VCO	Paraffin oil	FTIR-ATR (4000–400 cm^−1^)	Qualitative analysis: LDA, PCA Quantitative analysis: PCR, PLSR PLSR-R^2^: 0.999, RMSEC: 0.142 PCR-R^2^: 0.998, RMSEC: 0.204	[90]
VCO	Corn oil and sunflower oil	FTMIR (4000–650 cm^−1^)	PLS Corn oil-R^2^: 0.999, RMSEC: 0.866 Sunflower oil-R^2^: 0.999, RMSEC: 0.374	[75]
VCO	Canola oil	FTIR (4000–650 cm^−1^)	PLS, PCR and DA PLS-R^2^: 0.998, RMSEC: 0.392 PCR-R^2^: 0.990, RMSEC: 1.37	[91]
VCO	Palm kernel olein	FTIR (4000–650 cm^−1^)	PLS and DS R^2^: 0.9875, RMSECV: 1.70	[92]
VCO	Olive oil and palm oil (binary mixture)	FTIR (4000–650 cm^−1^)	PLS and PCR PLS: VCO in OO-R^2^: 0.9992, RMSEC: 0.765 VCO in PO-R^2^: 0.9996, RMSEC: 0.494 PCR: VCO in OO-R^2^: 0.9991, RMSEC: 0.768 VCO in PO-R^2^: 0.9742, RMSEC: 3.86	[46]
VCO	Palm oil and olive oil (ternary mixture)	FTIR (4000–650 cm^−1^)	PLS (2nd derivative) R^2^: 0.999, RMSEC: 0.200 PCR: R^2^: 0.999, RMSEC: 1.30	[75]
VCO	Lard	FTIR (4000–650 cm^−1^)	PLS and DA PLS: R^2^: 0.9990, RMSEC:0.722	[93]
VCO	Grape seed oil, soybean oil	FTIR (4000–650 cm^−1^)	PLS: GSO in VCO-R^2^: 0.998, RMSEC:0.007 VCO in SO-R^2^: 0.999, RMSEC: 0.268 PCR: GSO in VCO-R^2^: 0.998, RMSEC:0.622 VCO in SO-R^2^: 0.999, RMSEC:0.208	[67]
Coconut water	Sucrose, glucoseand fructose	1D proton NMR spectroscopy	PLSR (combined region) R^2^: 0.999, RMSEC:0.5889	[94]
Coconut water	Sucrose, glucose, fructose and high-fructose corn syrup (HFCS)	RS	PLSR Sucrose: R^2^: 0.9997, RMSEC:1.1551 Glucose: R^2^: 0.9997 RMSEC:1.3020 fructose: R^2^: 0.9996 RMSEC:1.3105 HFCS: R^2^: 0.9998 RMSEC:4.2641	[95]
Coconut water	sugar	IRMS coupled with an elemental analyzer		[96]

**Table 4 molecules-27-03250-t004:** Application of spectroscopic techniques for quantifying heavy metals in edible coconut products.

Coconut Product	Spectral Analysis	Salient Findings	Reference
Fresh coconut Coconut milk, cream, and milk powder	AAS	Fresh coconut showed greater contents of Fe, Cu, and Mn Milk contained more Zn	[43]
Coconut water Coconut milk	AAS	Fe > Zn > Cu > Pb Coconut water showed lower concentrations of Fe, Pb, Cu, and Zn than milk	[96]
VCO	AAS	Metal contents changed with the extraction process Fermentation process resulted in greater contents of Cu and Fe	[78]
Fermented coconut oil	AAS	Fermented oil showed less heavy metal chelating activity than traditional and commercially available oils	[105]
Coconut water	ICP-AAS	Higher content of Pb in mature coconut water Pb content in tender coconut water within acceptable limits	[105]
Coconut water Coconut milk	High-resolution continuum source graphite FAAS	Method presented appropriate precision, accuracy, and LoD Cd and Pb contents were below the maximum permissible limits	[102,103]
Coconut water	AAS	Concentrations of Ni, Cd, Cr, and Pb exceeded the toxicity level Low concentration of essential nutrients	[96]

## Data Availability

Data are available from the authors R.P. and R.K.

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
