# Peer review of "Contemporary Developments and Emerging Trends in the Application of Spectroscopy Techniques: A Particular Reference to Coconut (Cocos nucifera L.)"

_molecules, 2022, doi:10.3390/molecules27103250_

Round 1
Reviewer 1 Report
I consider this manuscript as a deep and rigorous revision about contemporary developments and emerging trends in the application of spectroscopy techniques. I recommend the paper to be considered for publication in Molecules. Nevertheless, there are two minor considerations that the authors should be clarified previously:
1.There are only 2.1 sections in the second chapter, and some section titles seem to be missing.
2.The detection of microbial contaminations is mentioned in the abstract. However, there seems to be no relevant content in the manuscripts.
Author Response
Reviewer 1
I consider this manuscript as a deep and rigorous revision about contemporary developments and emerging trends in the application of spectroscopy techniques. I recommend the paper to be considered for publication in Molecules. Nevertheless, there are two minor considerations that the authors should be clarified previously:
1.There are only 2.1 sections in the second chapter, and some section titles seem to be missing.
Thank you for the comments, the sections has been revised and structured accordingly.
2.The detection of microbial contaminations is mentioned in the abstract. However, there seems to be no relevant content in the manuscripts
The changes has been addressed.
Reviewer 2 Report
The manuscript under review highlights the exciting and important techniques for determining the quality attributes of coconut since the conventional techniques are destructive. He also could describe these advanced Spectroscopic methods and justified their utilization and how they distinctively evaluate various coconut products' quality characteristics to reach their authenticated consumer market. Although the abstract is informative and constrictive, it should include the objective of the review obviously.
Overall, the study is well designed and articulated. The write-up is acceptable, moreover, I think the authors followed the instructions and format of the Molecules journal, however, the funding and acknowledgment sections have not been written. In addition, although the article is of review type, it needs to be divided into many sections with different subheadings.
My decision would be to ACCEPT the article to be published in the Journal Molecules with minor revision.
Author Response
General comments
The authors present a review of the literature regarding current developments and emerging trends the analysis of Cocos nucifera using non-destructive analytical methods/equipment – specifically, spectroscopic techniques. The topic of discussion is relevant and merits review. The manuscript is sufficiently comprehensive, and the technical aspects of the topic are also well described/discussed. Nonetheless, the article contains numerous grammatical and typographical errors and needs English revision. Some specific comments on the manuscript can be found below.
Specific comments
The introductory part of the manuscript is disorganized and lacks logical flow. The authors should provide a more concise, and yet comprehensive introduction section. The rest of the information can be presented as sub-headings in the manuscript.
The introduction has been edited and structured accordingly.
Page 2, lines 62-63: Provide a reference for the sentence “Simultaneously, over 50% of the market for desiccated coconut powder is dominated by Indonesia and the Philippines, according to the Asian and Pacific Coconut Community (APCC), 2015”
The comment has been addressed and highlighted.
Page 2, line 75: Recast the phrase “…healthy/sports drink with numerous health benefits and is used as a rehydration drink…”
The comment has been addressed and highlighted.
Page 3, lines 78-79: Correct the sentence “Coconut meat, coconut milk, coconut cream, desiccated coconut powder, and coconut oil extract from coconut meat”
The comment has been addressed and highlighted.
Page 4, lines 146-147: Correct the sentence “…when a set of spectra is recorded at a different wavelength).
The comment has been addressed and highlighted.
Page 4, lines 159-160: Correct the sentence “During the photon and sample, both elastic and inelastic scattering occurs”
The comment has been addressed and highlighted.
Page 4, lines 175-178: Correct the sentence “The NIR spectra deal with the overtones' responses and higher vibrations of the molecular bonds, which cause lesser absorptivity of NIR light in the organic materials, but the incidence of these forbidden transitions renders this region unique distinct from other methods”
The comment has been addressed and highlighted.
Page 6, line 246: Considering the position of Table 1 in the manuscript, it appears the references in the table do not follow the sequential reference numbering of the rest of the article.
Thank you, the references in the table is formatted according to the reference at the end of the manuscript which correctly presented.
Page 7, lines 249-250: Correct the sentence “It was concluded that NMR was an effective method to analyze fats' constituents and differentiate the oils.[43] have characterized the lipid profile of coconut oils by…”
The comment has been addressed and highlighted.
Page 10, lines 338-339: The word “compassionate” in the sentence “It was a compassionate tool to monitor coconut water deterioration during the different stages of maturity” should be replaced.
The comment has been addressed and highlighted.
Page 10, line 368: Correct the sentence “FTIR was studied to evaluate the biochemical changes in VCO during thermal oxidation”
The comment has been addressed and highlighted.
Page 11, line 385: Correct this reference citation in the manuscript “Zicker, Craig, de Oliveira Ramiro, Franca, Labanca and Ferreira [72]”
The comment has been addressed and highlighted.
Page 14, lines 502-504: Correct the sentence “Applying the chemometric technique of PLS and Discriminant analysis (DA) was later used to determine the canola oil's adulteration in VCO quantitatively”
The comment has been addressed and highlighted.
Page 14, line 508: Correct the word “chemo-metric”
The comment has been addressed and highlighted.
Page 17, line 628: Correct the reference citation style in this line “was analyzed using AAS.67”
The comment has been addressed and highlighted.
Reviewer 3 Report
General comments
The authors present a review of the literature regarding current developments and emerging trends the analysis of Cocos nucifera using non-destructive analytical methods/equipment – specifically, spectroscopic techniques. The topic of discussion is relevant and merits review. The manuscript is sufficiently comprehensive, and the technical aspects of the topic are also well described/discussed. Nonetheless, the article contains numerous grammatical and typographical errors and needs English revision. Some specific comments on the manuscript can be found below.
Specific comments
The introductory part of the manuscript is disorganized and lacks logical flow. The authors should provide a more concise, and yet comprehensive introduction section. The rest of the information can be presented as sub-headings in the manuscript.
Page 2, lines 62-63: Provide a reference for the sentence “Simultaneously, over 50% of the market for desiccated coconut powder is dominated by Indonesia and the Philippines, according to the Asian and Pacific Coconut Community (APCC), 2015”
Page 2, line 75: Recast the phrase “…healthy/sports drink with numerous health benefits and is used as a rehydration drink…”
Page 3, lines 78-79: Correct the sentence “Coconut meat, coconut milk, coconut cream, desiccated coconut powder, and coconut oil extract from coconut meat”
Page 4, lines 146-147: Correct the sentence “…when a set of spectra is recorded at a different wavelength).
Page 4, lines 159-160: Correct the sentence “During the photon and sample, both elastic and inelastic scattering occurs”
Page 4, lines 175-178: Correct the sentence “The NIR spectra deal with the overtones' responses and higher vibrations of the molecular bonds, which cause lesser absorptivity of NIR light in the organic materials, but the incidence of these forbidden transitions renders this region unique distinct from other methods”
Page 6, line 246: Considering the position of Table 1 in the manuscript, it appears the references in the table do not follow the sequential reference numbering of the rest of the article.
Page 7, lines 249-250: Correct the sentence “It was concluded that NMR was an effective method to analyze fats' constituents and differentiate the oils.[43] have characterized the lipid profile of coconut oils by…”
Page 10, lines 338-339: The word “compassionate” in the sentence “It was a compassionate tool to monitor coconut water deterioration during the different stages of maturity” should be replaced.
Page 10, line 368: Correct the sentence “FTIR was studied to evaluate the biochemical changes in VCO during thermal oxidation”
Page 11, line 385: Correct this reference citation in the manuscript “Zicker, Craig, de Oliveira Ramiro, Franca, Labanca and Ferreira [72]”
Page 14, lines 502-504: Correct the sentence “Applying the chemometric technique of PLS and Discriminant analysis (DA) was later used to determine the canola oil's adulteration in VCO quantitatively”
Page 14, line 508: Correct the word “chemo-metric”
Page 17, line 628: Correct the reference citation style in this line “was analyzed using AAS.67”
Author Response
Comments and Suggestions for Authors
In my opinion manuscript molecules-1709415 could be published after major revision. The subject is interesting but the PRISMA guidelines (see BMJ 2021:372:n71 DOI: 10.1136/bmj.n71) should be followed closely in writing. At this point the manuscript is difficult to follow and should be better structured.
Suggestion to improve the manuscript:
- In the abstract the authors should clearly state why the review was done, what was done and what was found (see please Table 2 at page 5 of the PRISMA 2020 guidelines, BMJ 2021). Mention the method used, provide the results in short and discuss briefly the results and their interpretation.
Thank you, the comments are addressed as per the reviewer suggestion.
- In the Introduction indicate the rationale and the objectives (see please table 1 of the PRISMA 2020 paper, table 1 at pg 4)
Thank you. The introduction part have been modified as per the comments. Eventhough few lines were already used in the introduction section to explain the rationale (line:112-121), another section have been added to the end of introduction to explain the rationale and objectives.
- Introduce the methods section as described by the PRISMA guidelines.
Materials section as per the prisma guidelines are included in the manuscript from line no: 265- 275.
- Rename section 2 Results and add subsection 2.1 Application of spectroscopy techniques for quality evaluation
2.1.1. coconut oil
2.1.2. coconut milk
etc or use a different organizing system.
The suggestion have been made in the manuscript in Line: 276, 277,284, 301, 336, 386, 436, 469, 507, 513, 533, 610, and 643
- The novelty of the review should be emphasized at the end of the Conclusions.
The comment have been addressed in “the current review…..coconut based products”.
- references 13 and 14 are incomplete. about 30% of references are older than 10 years. if possible some recent studies should be considered also.
The references have been updated and new references were added.
Reviewer 4 Report
In my opinion manuscript molecules-1709415 could be published after major revision. The subject is interesting but the PRISMA guidelines (see BMJ 2021:372:n71 DOI: 10.1136/bmj.n71) should be followed closely in writing. At this point the manuscript is difficult to follow and should be better structured.
Suggestion to improve the manuscript:
- In the abstract the authors should clearly state why the review was done, what was done and what was found (see please Table 2 at page 5 of the PRISMA 2020 guidelines, BMJ 2021). Mention the method used, provide the results in short and discuss briefly the results and their interpretation.
- In the Introduction indicate the rationale and the objectives (see please table 1 of the PRISMA 2020 paper, table 1 at pg 4)
- Introduce the methods section as described by the PRISMA guidelines.
- Rename section 2 Results and add subsection 2.1 Application of spectroscopy techniques for quality evaluation
2.1.1. coconut oil
2.1.2. coconut milk
etc or use a different organizing system.
- The novelty of the review should be emphasized at the end of the Conclusions.
- references 13 and 14 are incomplete. about 30% of references are older than 10 years. if possible some recent studies should be considered also.
Author Response
Comments and Suggestions for Authors
In my opinion manuscript molecules-1709415 could be published after major revision. The subject is interesting but the PRISMA guidelines (see BMJ 2021:372:n71 DOI: 10.1136/bmj.n71) should be followed closely in writing. At this point the manuscript is difficult to follow and should be better structured.
Suggestion to improve the manuscript:
- In the abstract the authors should clearly state why the review was done, what was done and what was found (see please Table 2 at page 5 of the PRISMA 2020 guidelines, BMJ 2021). Mention the method used, provide the results in short and discuss briefly the results and their interpretation.
Thank you, the comments are addressed as per the reviewer suggestion.
- In the Introduction indicate the rationale and the objectives (see please table 1 of the PRISMA 2020 paper, table 1 at pg 4)
Thank you. The introduction part have been modified as per the comments. Eventhough few lines were already used in the introduction section to explain the rationale (line:112-121), another section have been added to the end of introduction to explain the rationale and objectives.
- Introduce the methods section as described by the PRISMA guidelines.
Materials section as per the prisma guidelines are included in the manuscript from line no: 265- 275.
- Rename section 2 Results and add subsection 2.1 Application of spectroscopy techniques for quality evaluation
2.1.1. coconut oil
2.1.2. coconut milk
etc or use a different organizing system.
The suggestion havebeen made in the manuscript in Line: 276, 277,284, 301, 336, 386, 436, 469, 507, 513, 533, 610, and 643
- The novelty of the review should be emphasized at the end of the Conclusions.
The comment have been addressedin “the current review…..coconut based products”.
- references 13 and 14 are incomplete. about 30% of references are older than 10 years. if possible some recent studies should be considered also.
The references have been updated and new references were added.
Round 2
Reviewer 4 Report
Paper molecules-1709415 was improved after revision and merits publication.